# Cerebellar Intermittent Theta-Burst Stimulation and Motor Control Training in Individuals with Cervical Dystonia

**DOI:** 10.3390/brainsci6040056

**Published:** 2016-11-23

**Authors:** Lynley V. Bradnam, Michelle N. McDonnell, Michael C. Ridding

**Affiliations:** 1Discipline of Physiotherapy, Graduate School of Health, University of Technology Sydney, Sydney, NSW 2007 Australia; 2Discipline of Physiotherapy, School of Health Sciences, Flinders University, Adelaide, SA 5001, Australia; 3Sansom Institute for Health Research, School of Health Sciences, University of South Australia, Adelaide, SA 5001, Australia; michelle.mcdonnell@unisa.edu.au; 4Robinson Research Institute, School of Medicine, University of Adelaide, Adelaide, SA 5005, Australia; michael.ridding@adelaide.edu.au

**Keywords:** cerebellum, neuromodulation, cervical dystonia, TWSTRS, CDQ-24

## Abstract

Background: There is emerging evidence that cervical dystonia is a neural network disorder with the cerebellum as a key node. The cerebellum may provide a target for neuromodulation as a therapeutic intervention in cervical dystonia. Objective: This study aimed to assess effects of intermittent theta-burst stimulation of the cerebellum on dystonia symptoms, quality of life, hand motor dexterity and cortical neurophysiology using transcranial magnetic stimulation. Methods: Sixteen participants with cervical dystonia were randomised into real or sham stimulation groups. Cerebellar neuromodulation was combined with motor training for the neck and an implicit learning task. The intervention was delivered over 10 working days. Outcome measures included dystonia severity and pain, quality of life, hand dexterity, and motor-evoked potentials and cortical silent periods recorded from upper trapezius muscles. Assessments were taken at baseline and after 5 and 10 days, with quality of life also measured 4 and 12 weeks later. Results: Intermittent theta-burst stimulation improved dystonia severity (Day 5, −5.44 points; *p* = 0.012; Day 10, −4.6 points; *p* = 0.025), however, effect sizes were small. Quality of life also improved (Day 5, −10.6 points, *p* = 0.012; Day 10, −8.6 points, *p* = 0.036; Week 4, −12.5 points, *p* = 0.036; Week 12, −12.4 points, *p* = 0.025), with medium or large effect sizes. There was a reduction in time to complete the pegboard task pre to post intervention (both *p* < 0.008). Cortical neurophysiology was unchanged by cerebellar neuromodulation. Conclusion: Intermittent theta-burst stimulation of the cerebellum may improve cervical dystonia symptoms, upper limb motor control and quality of life. The mechanism likely involves promoting neuroplasticity in the cerebellum although the neurophysiology remains to be elucidated. Cerebellar neuromodulation may have potential as a novel treatment intervention for cervical dystonia, although larger confirmatory studies are required.

## 1. Introduction

Cervical dystonia (CD) is a neurological disorder characterised by involuntary muscle contractions causing intermittent or sustained abnormal postures and/or tremor of the neck [1,2]. Cervical dystonia is debilitating, painful and strongly associated with depression and reduced quality of life [3,4]. As the contributing brain pathophysiology is unclear, the primary course of treatment is repeated botulinum toxin injections into affected muscles. There is a pressing need to identify new treatment interventions that modulate brain neuroplasticity to target aberrant mechanisms resulting in CD.

Recent evidence suggests the cerebellum is involved in the pathophysiology of CD [5,6,7]. Altered Purkinje cell structure, evidenced by thinner dendrites and fewer dendritic spines, was observed in a dystonic mouse model along with Purkinje cell displacement into the molecular layer [8]. Surgical removal of the cerebellum abolished abnormal movements in a rat model presenting with generalized dystonia [9]. Another dystonic mouse model developed using conditional genetics to regionally limit cerebellar dysfunction revealed Purkinje cell abnormalities causing dystonia-like postures [10] while progressive loss of Purkinje cell neurons reduced dystonia severity [11]. Further studies using dystonic rodent models localised the mechanism of action to Purkinje cell sodium pump dysfunction, leading to aberrant high frequency firing [12,13]. There is also accumulating evidence of cerebellar involvement in CD in humans. Neuropathological evidence from post mortem dystonic human brains indicates pathology within the cerebellar cortex and Purkinje cells [5]. Neuroimaging studies revealed abnormalities of grey and white matter in the cerebellum in various presentations of dystonia, implicating defects in the cerebello-thalamo-cortical pathway [14,15,16,17,18,19]. Neurophysiological experiments in humans concur. Participants with CD and focal hand dystonia lack usual adaptation to the cerebellar-dependent conditioned blink reflex [20], while non-invasive stimulation of the cerebellum normalised eye blink conditioning in focal hand dystonia [21]. Transcranial magnetic stimulation (TMS) studies revealed reduced modulation of sensorimotor plasticity, assessed by paired associative stimulation, following repeated TMS over the cerebellum in individuals with focal hand dystonia [22]. The accumulating evidence has led researchers to consider dystonia as a network disorder with the cerebellum as a key node [5,6,7,15,19,23,24,25,26,27,28,29].

One clinical translational possibility arises; the cerebellum may provide a therapeutic target for neuromodulation in neurological diseases associated with cerebellar dysfunction [30,31,32,33,34]. Because of its superficial location at the posterior skull, the cerebellar cortex can be modulated by non-invasive brain stimulation [35] to evoke neuroplasticity. The putative effect is to influence excitability of cerebello-thalamo-cortical [36,37] and brainstem output pathways. Repetitive TMS protocols in the form of theta-burst stimulation (TBS) can be delivered in two patterns. Intermittent TBS (iTBS) increases neuronal excitability and continuous TBS (cTBS) has a suppressive effect [38,39,40]. Evidence from a positron emission tomography (PET) study established that cTBS modulated neuronal activity in the cerebellar cortex and dentate nucleus in individuals with Parkinson’s disease [31]. In CD, 10 sessions of cerebellar cTBS produced a clinical improvement [41]. The effect of inducing neuroplasticity in the cerebellum in CD using iTBS has not yet been examined. This is important to understand, as a similar method of cerebellar neuromodulation—anodal direct current stimulation—was found to improve handwriting in focal hand dystonia [42]. The primary aim of this study was to test the effect of cerebellar iTBS on dystonia severity in individuals with CD. The secondary outcome measures were quality of life and motor coordination, assessed by hand dexterity and TMS to assess neuroplasticity of M1 upper trapezius (UT) muscle representations, a muscle commonly dystonic in CD. We hypothesised that (i) dystonia severity would be reduced by iTBS; (ii) quality of life and hand dexterity would improve and (iii) cerebellar neuromodulation would influence M1 excitatory and inhibitory circuits assessed using TMS.

## 2. Experimental Section

### 2.1. Participants

Sixteen participants with CD diagnosed by a neurologist were enrolled in the study (age range 28–72 years, 6 males). Five participants were not receiving botulinum toxin injections at the time of the study, nor had they received botulinum injections in the prior six months. The remaining 11 participants who received regular botulinum toxin were at least 10 weeks post injection at the time of the study, verified by the neurology appointment card. All participants were screened for safety to undergo TMS, using a customised version of a standard tool [43] by a medical doctor prior to enrolment. Each provided written informed consent. All procedures were approved by the Southern Adelaide Clinical Human Research Ethics Committee (SAC HREC 411.12) and the trial was registered on the Australian New Zealand Clinical Trials Registry (ACTRN12612001182886).

### 2.2. Experimental Design

Participants were randomised into real or sham iTBS groups using a minimisation procedure [44] taking into account age, sex, Toronto Western Spasmodic Rating Scale (TWSTRS) [45] scores and current treatment with botulinum toxin injections, to ensure comparable groups at baseline. Participants attended the laboratory for 12 sessions. Baseline (PRE) and immediate post (POST 1) intervention outcomes were assessed in the first and last sessions. Sessions 2–11 consisted of the experimental interventions, delivered at the same time on ten consecutive working days by an independent investigator. Mid-intervention (MID) outcomes were assessed at day 7, prior to delivery of the intervention, with a one-hour break between TMS assessments and cerebellar stimulation. The experimental protocol is provided in Figure 1. Investigators collecting the neurophysiological and behavioural outcome measures were blinded to group allocation. Participants were also blinded as to whether they were allocated to the real or sham iTBS groups until completion of the study.

### 2.3. Electromyography (EMG)

Surface EMG was recorded from the UT muscles bilaterally using 10-mm diameter Ag/AgCl electrodes (Ambu, Ballerup, Denmark). The UT was chosen as it is commonly affected in CD and EMG can be recorded using surface electrodes. Electrodes were placed over the muscle on the high point of the shoulder, mid-way between the vertebral column and the acromion process (Figure 2). A 20 mm-diameter reference Ag/AgCl electrode was placed over the spinous process of C7 (3 M Health Care, St. Paul, MN, USA). EMG signals were sampled at 2000 Hz (CED 1401; Cambridge Electronic Design, Cambridge, UK), amplified by a factor of 1000 (CED 1902; Cambridge Electronic Design, Cambridge, UK), band-pass filtered (20–1000 Hz) and stored for offline analysis (Signal v5.09, Cambridge Electronic Design, Cambridge, UK).

### 2.4. Transcranial Magnetic Stimulation

Single-pulse TMS was delivered with a figure of eight coil (70 mm wing diameter, MagStim Co., Whitland, UK), positioned over M1 with the handle pointing backwards and laterally, to induce a posterior-to-anterior directed electrical current in the brain. The hotspots for evoking contralateral UT motor-evoked potentials (MEPs) and cortical silent periods (CSPs) were located and marked on the scalp. To pre-activate the muscle for TMS, the participant was asked to lift the arm into a position of shoulder flexion at approximately 45 degrees in the sagittal plane with the elbow extended and forearm and wrist in neutral [46]. A schematic illustration of the TMS coil, electrode placement and arm position is provided in Figure 2. This position consistently produced a contraction of approximately 20% of a maximal voluntary contraction in pilot testing. The participant was trained in the arm elevation task prior to data collection and visual feedback of EMG was provided to assist task performance. The average background EMG activity in the UT during this task was calculated for each individual at the first session. At the MID and POST 1 sessions, EMG activity was carefully monitored by the experimenters and displayed on the computer screen for participants to see during the experiment. If the contraction deviated by greater than ≈2 standard deviations (SD) from the baseline mean, the participant was verbally encouraged to increase or decrease the muscle contraction.

Active motor threshold (AMT) was determined as the minimum stimulus intensity to elicit a MEP of at least 100 µV in at least four out of eight trials, during the same task [47]. Sixteen TMS pulses at 130% AMT were applied at a rate of 0.2 Hz to the hotspots for the UT muscles contralateral and ipsilateral to the direction of dystonic head turn, with the contralateral arm held in the task position. Data were collected in blocks of eight TMS pulses, with extra rest breaks provided if required due to muscle fatigue or pain.

### 2.5. Theta-Burst Stimulation

Intermittent TBS was delivered by a Rapid 2 Magnetic stimulator (Magstim Co. Ltd.) via a 70 mm flat figure of eight coil using the protocol described by Huang et al., a 2 s train of TBS repeated every 10 s for a total of 190 s or 600 pulses [40]. The coil was centred over the lateral cerebellum, 3 cm lateral and 1 cm inferior to the inion with the handle pointing superiorly [36,48]. Sham iTBS was applied using an identical sham TMS coil (Magstim Co. Ltd.) which delivered a similar acoustic noise to that of real iTBS, but no stimulation was delivered to the brain. Due to the complex nature of the pattern of cervical muscles involved in CD, the target cerebellar hemisphere was uncertain. Therefore, we stimulated the cerebellum bilaterally conforming to a previous protocol in Parkinson’s disease patients [41]. The lateral cerebellar cortices were stimulated in a pseudo-randomised order for each session, with a 2-min rest between sides so each iTBS session took about 500 s to deliver. Participants sat quietly for 5 min after stimulation and then performed active exercise training guided by a video recording. One of two ten-minute videos was delivered randomly at each session by the same investigator who performed iTBS. The videos depicted a person sitting in a chair performing neck exercises with voice-over instructions. One video guided participants to perform active exercises out of the dystonic position, while attempting to maintain a ‘chin tuck’ neutral upper neck posture. This was followed by action observation training, where the model performed a sequence of five movements and participants were asked to observe, remember and physically repeat the sequence. The second video began with active exercises, followed by a ‘cueing task’, where participants were asked to move out of their dystonic posture in response to a sound cue. The motor control training was selected to encourage ‘motor sequencing’, a deficit known to rely on intact cerebellar function [49,50]. Participants were instructed to ‘just do the best they could’ and could stop or rest at any time in order to keep the exercise within their tolerance level. All participants had limited cervical range of motion and could only perform the exercises in their usual range. The videos were followed by three blocks of laterality recognition (implicit learning) training delivered by a computer program (Recognise, NOI, Adelaide, Australia). The implicit motor learning task was also included to activate the cerebellum as implicit learning is a cerebellar function [51,52]. Twenty images of a body part (hands, feet, knees or shoulders, chosen at random) were shown every 15 s in each block with a short rest between blocks. Participants were asked to press the right or left hand arrow on the computer keyboard as fast as possible to indicate the side of the body shown for each image. Accuracy and time to response were recorded in a personal diary so participants could track progress and were encouraged to improve scores at each session. All participants easily completed the three blocks at each training session.

### 2.6. Questionnaires and Grooved Pegboard Task

The TWSTRS is a validated assessment scale used to measure the negative impact of CD [45]. It is comprised of three subscales: Severity, Disability and Pain, with each subscale scored independently. The total TWSTRS score ranges from 0 to 87, with higher scores indicating higher negative impact. The Craniocervical Dystonia Questionnaire 24 (CDQ-24) is a validated, self-reported, CD-specific quality of life measure [53]. The 24-item CDQ-24 consists of five subscales: Stigma, Emotional wellbeing, Pain, Activities of daily living, and Social/family life. Scores range from 0 to 96, with higher scores indicating reduced quality of life. The disability and pain subsections of the TWSTRS and the CDQ-24 were completed by the participant at each data collection point. The severity component was assessed via a video recording by an investigator (MM) who was blinded to group allocation and session order. Hand dexterity as a measure of motor coordination was tested using a grooved pegboard task (Model 32025, Lafayette Instruments). This measure was included as evidence that the cerebellum was modulated in a favourable manner by iTBS. Participants were timed (seconds) as they placed 25 pegs into grooved holes positioned in a 5 × 5 square grid at different orientations with their non-dominant hand in a single trial. The non-dominant hand was chosen to maximise task difficulty.

### 2.7. Data Analysis

Data were checked for normality using the Shapiro-Wilks test. Baseline data were compared between groups using Mann-Whitney U tests or independent samples *t*-tests. Questionnaires (ordinal data) of total TWSTRS score, TWSTRS pain subscale and CDQ-24 were analysed using separate non-parametric Friedman’s tests. Significant within-group differences from baseline were tested using Wilcoxon signed rank post hoc tests and the difference between groups examined with Mann-Whitney U tests. Grooved pegboard task data were analysed with a 2 GROUP (real, sham) by 3 time (PRE, MID, POST 1) repeated measures ANOVA. Post hoc effects were explored by paired t-tests. Effect sizes at MID and POST 1 from baseline were calculated for each behavioural measure using the Cohen’s *d* statistic, with 0.2, 0.5 and 0.8 indicating small, medium and large effect sizes respectively [54].

For MEP analysis, EMG data were rectified off line using Signal Software (Signal v5.09, Cambridge Electronic Design, Cambridge, UK), and MEP area (mV·S) calculated using the same window for all traces for each individual and then averaged [55,56,57]. The CSP duration (ms) was calculated from the same EMG traces [57]. The onset was determined as the point where rectified EMG activity dropped below the prestimulus mean for at least 10 ms and the offset as the point where EMG activity permanently returned to the prestimulus baseline average [57,58]. The duration was measured from the stimulus artefact to the offset [58]. Each trace was visually inspected and measured for each individual. MEP and CSP data were normalized to baseline prior to statistical analysis (post/pre). Prestimulus root mean square EMG (rmsEMG) was measured for 100 ms prior to the stimulus artefact. Traces where background rmsEMG was outside of 2 SD from the mean for each individual at each session were discarded from analysis. No more than four traces per subject were discarded on this basis. The rmsEMG data did not meet assumptions of normality and were logarithmically transformed prior to statistical analysis. MEP area and CSP duration were analysed separately for each muscle (contralateral or ipsilateral UT) using two-group (real, sham), three time (PRE, MID, POST 1) repeated measures ANOVA. To ensure the muscle contraction during TMS measures was consistent across time, the prestimulus rmsEMG was analysed using a two-group (real, sham), two-side (contralateral, ipsilateral), three time (PRE, MID, POST 1) repeated measures ANOVA. SPSS software (Version 20, SPSS Inc., Chicago, IL, USA) was used for statistical analysis, with the level of significance set to *p* < 0.05. Mauchly’s test examined data for sphericity and the Greenhouse-Geisser correction was used where data were non-spherical. Post hoc corrections were applied using a modified Bonferroni test [59]. Data are presented as the groups mean ± standard error.

## 3. Results

Participant demographics are presented in Table 1.

There were no adverse effects of iTBS reported. There was no difference between the real and sham iTBS groups at baseline for age (*p* = 0.64), time since CD onset (*p* = 0.57) or behavioural data (all *p* > 0.56). The primary outcome measure, total TWSTRS score, was reduced by real iTBS (Χ^2^ (2) = 9.75, *p* = 0.008), but not sham iTBS (Χ^2^ (2) = 0.45, *p* = 0.79) (Figure 3A). Wilcoxon post hoc tests revealed the TWSTRS was reduced from baseline following real iTBS at MID (−5.44 points; *p* = 0.012) and POST 1 (−4.6 points; *p* = 0.025). There was no difference between groups at either time point (all *p* > 0.26). There was a reduction in the pain subsection of the TWSTRS following real iTBS (Χ^2^ (2) = 9.74, *p* = 0.008), but not sham iTBS (Χ^2^ (2) = 5.56, *p* = 0.068) (Figure 3B). Wilcoxon post hoc tests revealed a reduction in pain at MID (−2.94 points; *p* = 0.017) and POST 1 (−2.97 points; *p* = 0.04) following real iTBS. Group comparison revealed a difference at POST 1 (*p* = 0.04), but not at PRE or MID (both *p* > 0.17).

There was a reduction in the CDQ-24 after iTBS (Χ^2^ (4) = 10.17, *p* = 0.038), but not sham TBS (Χ^2^ (4) = 4.38, *p* = 0.54) (Figure 3C). Post hoc tests revealed the CDQ-24 was reduced by iTBS compared to baseline (MID −10.6 points, *p* = 0.012; POST 1 −8.6 points, *p* = 0.036; POST 2 −12.5 points, *p* = 0.036; POST 3 −12.4 points, *p* = 0.025). Group comparison revealed a difference between iTBS and sham TBS at MID (*p* = 0.021), POST 2 (*p* = 0.014), and POST 3 (*p* = 0.007), but not POST 1 (*p* = 0.13).

For the grooved pegboard task there was a main effect of time (F_2,14_ = 6.22, *p* = 0.012) and a condition by time interaction (F_2,14_ = 4.45, *p* = 0.032) (Figure 4). Post hoc paired t-tests indicated there was a reduction in time to performing the task in the iTBS group from PRE to MID (−7.9 s, *p* = 0.005), from PRE to POST 1 (−15.8 s, *p* = 0.004) and from MID to POST 1 (−7.9 s, *p* = 0.008). There was no effect after sham iTBS (all *p* > 0.75). There was a difference between groups at POST 1 (*p* = 0.018), but not at PRE or MID (both *p* > 0.41). All effect sizes are provided in Table 2.

The raw data for MEP area and CSP duration are provided in Table 3 and normalised data in Figure 5. There were no main effects or an interaction for MEP area in ipsilateral UT (all *p* > 0.38) or contralateral UT (all *p* > 0.29). There were no main effects or an interaction for CSP duration in ipsilateral UT (all *p* > 0.21) or contralateral UT (all *p* > 0.33). There were no main effects or interactions for the logarithmic transformed rmsEMG for EMG activity in UT (all *p* > 0.12).

## 4. Discussion

The main findings were that modulating neuroplasticity of the cerebellum by ten daily sessions of cerebellar iTBS and motor and laterality training reduced pain and improved quality of life in individuals with CD. Motor coordination, assessed by hand dexterity, was also improved after real iTBS and training. There were no effects on MEP area or CSP duration; therefore, the mechanisms supporting the clinical and behavioural benefits remain uncertain. The results support the hypothesis that cerebellar iTBS has potential as novel a treatment intervention; although this must be interpreted conservatively due to the sample size until confirmed by further research.

Two studies now evidence that cerebellar TBS may induce beneficial neuroplasticity in people with CD. Both cerebellar iTBS in the current study and cerebellar cTBS in previous work by Koch and colleagues [41] reduced the total TWSTRS score by almost 5 points, a statistically significant decrease. A minimal clinical important difference (MCID) has not been calculated for the TWSTRS to date, so it is unclear if these point reductions are clinically relevant. To put the current findings in context, a 5-point reduction was also reported 3 months after pallidal deep brain stimulation [60] and after 12 weeks of physiotherapy motor retraining exercises [61]. However, a single case study of a CD patient reported an 18-point reduction in total TWSTRS score after 12 weeks of cerebellar anodal direct current stimulation in between botulinum toxin injections [62] and large scale trials of the efficacy of botulinum toxin injections report a 15-point average reduction in TWSTRS score 4 weeks after treatment [63,64,65]. There were three other main points of difference between the studies apart from the pattern of TBS stimulation. First, we used the self-reported CDQ-24 [49], a CD-specific quality of life measure, to determine immediate and longer term effects of cerebellar iTBS. There was a significant improvement in quality of life after the intervention period in the real cerebellar iTBS group, which continued across the 12 week follow up. The 11-point reduction in CDQ-24 score following treatment was comparable to a large clinical trial of botulinum toxin injections [63]. Second, we applied cerebellar iTBS followed by training exercises and implicit learning practice, rather than cerebellar stimulation alone. Interestingly, the training program was only effective when combined with cerebellar iTBS, as the sham iTBS group did not improve. A study is warranted to examine effects of iTBS in isolation to determine the relative contribution of cerebellar neuromodulation and motor control training. Third, we included a hand dexterity task to assess motor control, which improved in the real cerebellar iTBS group only. This latter finding supports modulation of the cerebellum by iTBS, which improved coordination and fine motor control of the hand. It is well known the cerebellum projects to the red nucleus and reticular nucleus in the brainstem as well as to M1. All three descending pathways have terminations in cervical motoneurons to control upper limb function which could explain this finding. Without comparable data from control subjects we cannot tell if there were deficits in hand dexterity in people with cervical dystonia at baseline. However, this study suggests cerebellar neuromodulation improves upper limb motor control in addition to dystonia severity in CD. Robust clinical trials are now needed to assess efficacy, cost effectiveness and patient preferences, along with determining a MCID for the TWSTRS so clinical effectiveness can be determined.

Cerebellar neuromodulation did not influence M1 excitability or inhibition in the current study. This is consistent with findings by Koch and colleagues [41], however the current study evoked responses from dystonic neck muscles while they probed non-dystonic hand muscles. Koch and colleagues also examined responses to paired associative stimuli and cerebellar-brain inhibition (CBI) [37,66,67] recorded from unaffected hand muscles [41]. Cerebellar cTBS modulated CBI and reduced the potentiation of responses to paired associative stimuli, indicating an influence on inhibitory circuits that influences M1 behaviour. Effects of cerebellar iTBS on CBI and paired associative stimulation recorded from dystonic neck muscles in people with CD remain to be elucidated.

It is uncertain why both cerebellar iTBS in the current study and cerebellar cTBS used by Koch and colleagues [41] produced a similar effect on TWSTRS scores. It may be the two patterns of stimulation engage different circuits within the cerebellar cortex to achieve a similar net effect, as cerebellar neuronal organisation is specialized and complex. Cerebellar Purkinje cells are modulated by excitatory inputs from climbing and parallel fibres and by inhibitory projections from basket and stellate cells. Dysfunction of these circuits in dystonia is supported by rodent models demonstrating dystonic muscle activation linked to aberrant intrinsic Purkinje cell firing, secondary to sodium pump and calcium channel deficits [12,68]. Cellular mechanisms of long-term potentiation (LTP) and depression (LTD) in the cerebellar cortex [69], in particular in calcium-mediated synaptic plasticity [70], may be affected. It is possible that TBS modulates LTP and LTD-like synaptic plasticity in the cerebellar cortex, similarly to the M1 [71,72]. Magnetic resonance spectroscopy may help to elucidate the mechanisms of action of TBS in the cerebellar cortex in humans.

The current results add to the growing body of evidence that the cerebellum is a key node in a network disorder likely involving cortical and subcortical brain [6,7,19,23,24,25,26]. How the cerebellum contributes to the dystonia phenotype remains uncertain. While speculative, it may be that cerebellar iTBS influences projections directly to brainstem nuclei involved in motor coordination of the neck and face, such as the red nucleus, superior colliculus or trigeminal nuclear complex [23]. Other authors have proposed models to explain the pathogenesis of CD, such as the interstitial nucleus of Cajal in the brainstem, responsible for integrating inputs from the cerebellum, vestibular nuclei and neck muscle proprioceptors, to control the position of the head [24,73,74]. Pontine reticulospinal neurons receiving monosynaptic excitatory input from the superior colliculus and inhibitory input from head-movement related omnipause neurons projecting to agonist and antagonist neck muscles to control head motion has also been proposed [75]. More experimental work is required to understand how cerebellar neuromodulation impacts on neural circuits subserving dystonia.

There were limitations to this study. First, iTBS was applied in combination with motor control training. Since there was no effect of sham iTBS and exercise, it appears the impact of motor training in isolation is minimal. The improvements noted could result from synergistic action of iTBS and exercise or by iTBS alone. Answering this question should be the focus of a future study. Second, the primary outcome measure TWSTRS was only assessed at the end of the intervention and not in the longer term follow up, so effects on dystonia severity beyond the intervention period are unknown. Third, we did not include a measure of cerebellar output, such as CBI, because CBI has not been yet been demonstrated to exist in UT cortical representations. However, TBS has consistently influenced hand muscle CBI [32,34,41], as has cerebellar direct current stimulation when the anode is placed over the lateral cerebellum [42,76]. Furthermore, PET scanning demonstrated robust metabolic effects in both cerebellar cortex and deep cerebellar nuclei following cTBS applied to the lateral cerebellum [31]. For these reasons we believe the cerebellum was modulated by iTBS in the current study. Recent studies applied ‘sham’ TBS over the posterior neck muscles rather than using sham coil over the cerebellum. A change to TMS measures evoked from M1 following cerebellar TBS with no difference after posterior neck TBS was taken as evidence of cerebellar neuromodulation [77,78]. This method could be utilised in a future study in CD. In the current study, TMS measures taken before and after the 10-session intervention period were unaffected. A future study could include TMS measures before and after a single session to probe short-term neuroplasticity responses in M1 as a result of cerebellar neuromodulation in people with CD as observed in Parkinson’s disease and essential tremor [77,78]. Finally, the small number of participants means the current findings must be replicated in larger numbers of patients.

## 5. Conclusions

Modulating neuroplasticity of the cerebellum using iTBS combined with exercise training has potential to improve CD, particularly for pain and quality of life. Cerebellar iTBS appears worthy of further investigation as a novel treatment for people living with CD.

## Figures and Tables

**Figure 1 brainsci-06-00056-f001:**
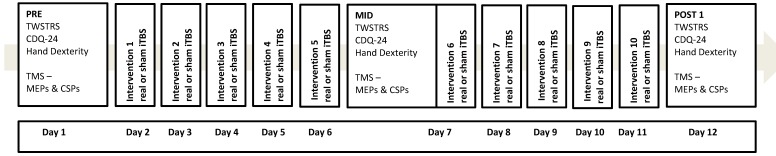
The experimental protocol showing the order of the experimental (PRE, MID, POST 1) and intervention sessions. Real or sham iTBS followed by motor control and laterality training was performed at each session. The MID experimental session and intervention session 6 were performed on the same day. Note: the CDQ-24 was additionally assessed 4 (POST 2) and 12 (POST 3) weeks after the intervention. TWSTRS: Toronto Western Spasmodic Rating Scale; CDQ-24: Craniocervical Dystonia Questionnaire24; TMS: transcranial magnetic stimulation, MEPs: motor-evoked potentials, CSP: cortical silent period.

**Figure 2 brainsci-06-00056-f002:**
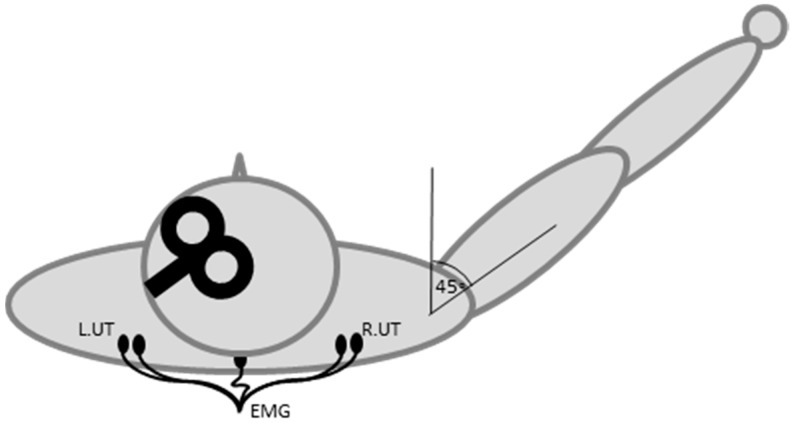
The TMS coil, EMG electrodes and arm position during the experiment.

**Figure 3 brainsci-06-00056-f003:**
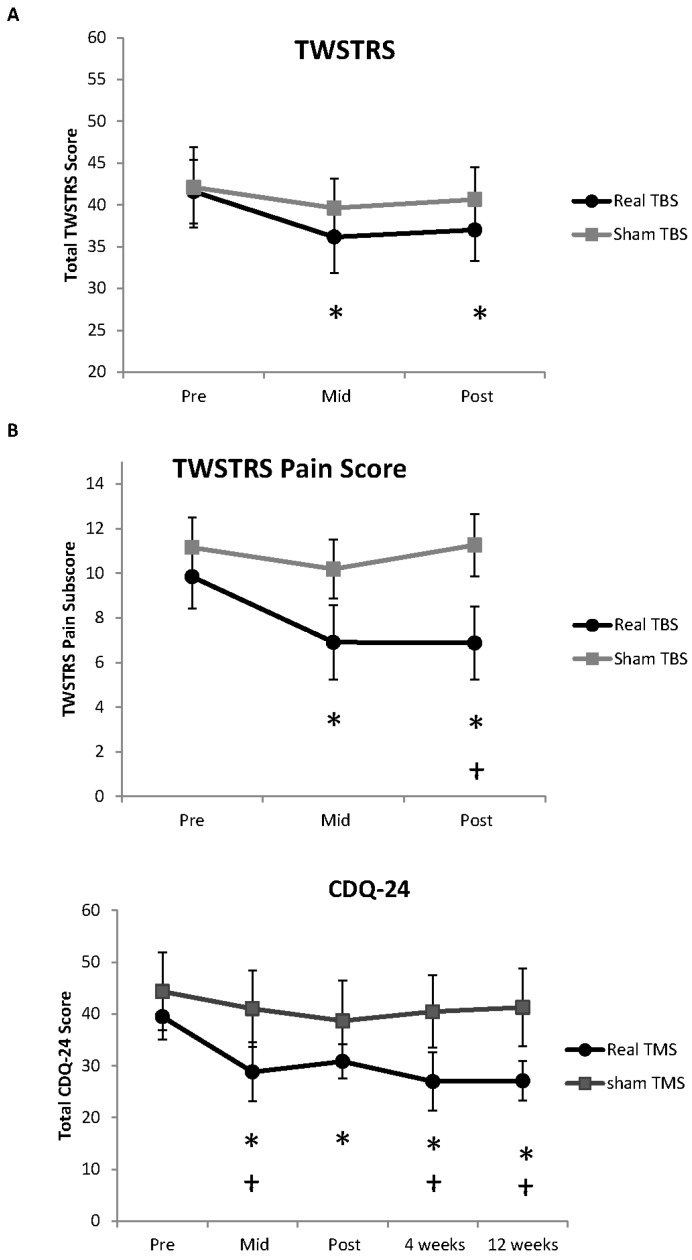
(**A**) Results for the TWSTRS. There was a reduction in the TWSTRS total score in the real iTBS group at MID and POST 1; (**B**) TWSTRS pain subscale. There was a reduction in the pain subscale in the real iTBS group at MID and POST 1, with a significant difference between groups at POST 1; (**C**) Results for the CDQ-24. There was a reduction in CDQ-24 at each time point compared to baseline in the real iTBS group. The difference between real and sham iTBS groups was significant at MID, POST 2 and POST 3. Significant within-group difference from baseline at *p* < 0.05 is signified by * and a significant between-group difference at *p* < 0.05 is signified by †.

**Figure 4 brainsci-06-00056-f004:**
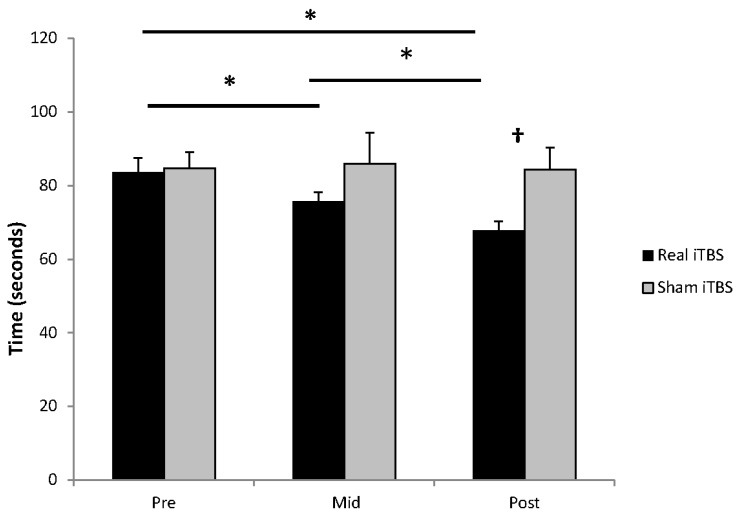
Effect of iTBS on the grooved pegboard task. There was a reduction in task time following real iTBS between PRE and MID, MID and POST 1 and between PRE and POST 1. There was no effect of sham iTBS. Significant difference from baseline in the real iTBS group at *p* < 0.05 is signified by * and a significant difference between the iTBS and sham iTBS groups at *p* < 0.05 is signified by †.

**Figure 5 brainsci-06-00056-f005:**
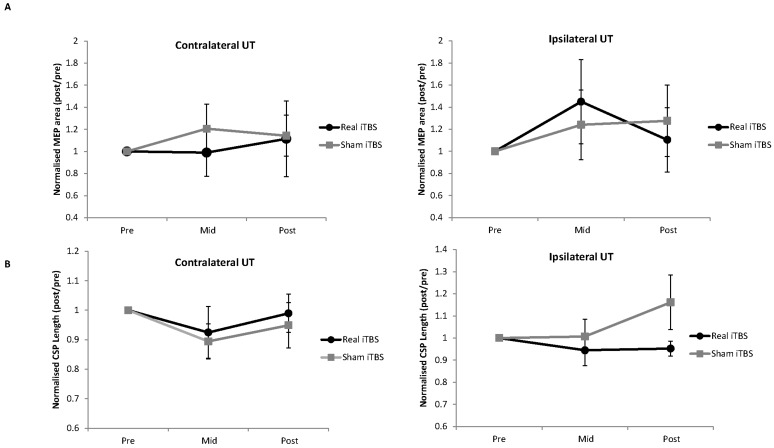
Effect of iTBS on TMS-evoked MEP area (**A**) and CSPs (**B**). Data were normalised to pre-intervention values. There were no main effects or interactions for either measure. UT = upper trapezius.

**Table 1 brainsci-06-00056-t001:** Participant characteristics.

Age (year)	Sex	Dystonia Type	Time since Dystonia onset (year)	TWSTRS 0-85	CDQ24 0-96	BTXN
**Real iTBS Group**						
46	F	Left Rotation	10	58.8	48.9	Yes
58	F	Right Rotation	15	40.3	31.3	No
63	F	Right Rotation	8	45.0	17.7	No
57	M	Right Rotation	3	30.3	33.3	Yes
42	F	Left Rotation/Right Side Flexion	4	48.3	44.8	No
38	M	Right Rotation	2	23.8	34.4	Yes
51	F	Right Rotation	4	42.5	54.1	Yes
49	M	Left Rotation	5	44.0	51.04	Yes
**Mean ± SD**						
50.5 ± 8.5			6.4 ± 4.4	41.6 ± 10.7	39.4 ± 12.4	
**Sham iTBS Group**						
72	M	Left Rotation	8	38.0	20.8	Yes
66	F	Left Rotation	5	47.5	50	No
43	M	Left SideFlexion/Flexion	2	44.5	57.3	Yes
62	M	Left Rotation	3	40.3	38.5	Yes
28	F	Right Rotation	5	26.3	13.5	Yes
62	F	Left Rotation	12	70.0	51.1	Yes
48	F	Left Rotation	3	27.5	42.7	Yes
46	F	Right Rotation	30	42.75	81.25	No
**Mean ± SD**						
53.4 ± 14.6			8.5 ± 9.3	42.1 ± 13.6	44.4 ± 21.21	

Abbreviations: TWSTRS: Toronto Western Spasmodic Rating Scale; CDQ-24: Craniocervical Dystonia Questionnaire-24. Higher scores indicate worse disorder severity or quality of life. BTXN = Botulinum toxin injections were standard treatment for these individuals.

**Table 2 brainsci-06-00056-t002:** Raw scores and effect sizes for clinical and behavioural measures in the real iTBS group.

	Pre	Mid	Effect Size	Post 1	Effect Size	Post 2	Effect Size	Post 3	Effect Size
TWSTRS	41.59 ± 3.8	36.12 ± 4.3	0.48 (S)	37.0 ± 3.7	0.43 (S)	N/A		N/A	
TWSTRS Pain	9.84375 ± 1.4	6.90625 ± 1.7	0.67 (M)	6.875 ± 1.6	0.68 (M)	N/A		N/A	
CDQ-24	39.45 ± 4.4	28.81 ± 4.3	0.74 (M)	30.84 ± 3.4	0.78 (M)	26.89 ± 5.7	0.88 (L)	27.09 ± 3.8	1.06 (L)
Pegboard (seconds)	83.63 ± 3.8	75.75 ± 2.4	0.87 (L)	67.88 ± 2.5	1.74 (L)	N/A		N/A	

TWSTRS = Toronto Western Spasmodic Torticollis Rating Scale, CDQ-24 = Craniocervical Dystonia Questionnaire, S = small, M = medium, L = large. Note effect sizes were calculated for within group difference from baseline in the real iTBS group.

**Table 3 brainsci-06-00056-t003:** Raw scores for MEParea and CSP duration for iTBS and sham iTBS.

	Contralateral Upper Trapezius			Ipsilateral Upper Trapezius		
**MEP area (mV.S)**	**Pre**	**Mid**	**Post**	**Pre**	**Mid**	**Post**
Real iTBS	1.85 ± 0.37	1.35 ± 0.21	1.34 ± 0.22	1.69 ± 0.17	2.15 ± 0.43	1.64 ± 0.30
Sham iTBS	1.53 ± 0.21	1.62 ± 0.23	1.66 ± 0.28	1.93 ± 0.42	1.84 ± 0.27	2.09 ± 0.44
**CSP length (ms)**	**Pre**	**Mid**	**Post**	**Pre**	**Mid**	**Post**
Real iTBS	95.71 ± 9.93	85.04 ± 8.02	90.79 ± 5.98	111.73 ± 7.64	105.81 ± 10.47	107.42 ± 10.25
Sham iTBS	90.04 ± 8.93	77.88 ± 5.72	81.44 ± 4.86	96.32 ± 9.63	93.56 ± 7.12	108.62 ± 12.83

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
