# Peer review of "Cerebellar Intermittent Theta-Burst Stimulation and Motor Control Training in Individuals with Cervical Dystonia"

_brainsci, 2016, doi:10.3390/brainsci6040056_

Round 1

Reviewer 1 Report

This manuscript is the first to combine a physical exercise program with cerebellar stimulation for those with CD. The main purpose was to assess changes in CD severity following excitatory cerebellar stimulation and neck exercises over 10 days. There was a significant improvement in pain and quality of life in the real group compared to sham. These results are encouraging that the cerebellum may be a promising target for therapeutic NIBS in dystonia.

This is a well written paper that makes an important addition to the field. There are a few issues that should be addressed however to strengthen the report.

Major:

Some of the scientific reasoning and methodology needs to be better clarified. Specifically,

1. Why perform hand dexterity in patients with CD?

2.  Why was iTBS hypothesized to begin with, particularly given the results of the cTBS study

3. There are no TMS data presented. This does a disservice to others that may want to replicate the methodology. Upper trap excitability measurement is not common and clearer methodology of it (including picture perhaps of coil location, describe ground location) should be added. The TMS data should be reported with values and even a trace would be helpful. How was % effort calculated?

4. The conclusion refers specifically to TMS to cerebellum as likely target for benefit, but the key factor of TMS combined with movement practice is left out of the conclusions and discussion. This must be discussed and the conclusions adapted accordingly.

5. Relatedly, please include better description of the intervention and clinical / scientific reasoning that went into the choice of those tasks. Eg, why the implicit task? Clarify duration of movement and tolerance of movement in the patients with CD.

6. There is some concern about the TMS processing. It is quite unconventional to report the MEP during a CSP, as slight variations in the EMG will have great influences on amplitude. The authors attempted to address this with an analysis of pre-stimulus activity, but this must be assessed in a trial by trial manner. Also, likely this analysis did not have enough power to show a difference. A suggestion would be determining a threshold of activity for each individual and excluding trials that exceed a given SD. Another thing that would make this methodology more robust would be to report the reliability within a subject across time. We know that CSP is highly reliable, but I am not familiar with reliability of an active contraction MEP, particularly in the upper trap. Overall, more information on this should be provided.

7. What was the justification of using area under curve vs. peak to peak? How was CSP onset/off set calculated? What was the normalization procedure?

Minor:

1.     An picture of the coil position would be helpful in describing it.

2.     An image of the upper trap TMS set up would be helpful.

3.     Specifically spell out the parameters of the stimulation. It is assumed from the writing that it is 2 sets of 10 min, but the timing in the paper refers to the video and not the stimulation parameters (line 151)

4.     Table 1 refers to Botox – should be BoTN for botulinum toxin

5.     TWSTRS is referred to as measure of function (Table 1 legend) but it is generally referred to as a measure of disease severity

6.     Typo line 195 – CSP measured in mm vs ms

7.     Consider adding in report of the other subscales in TWSTRS – call out pain, but not other ones

8.     Table 2 is confusing with the Post2, Post3 addition. I would recommend having more columns and just leaving some blank for measures that were not assessed at those time points, or have CDQ is different table.

9.     Use sex vs. gender

10.  Re; TMS analysis, was peak to peak analysis any different than AUC?

11.  Was the blind broken and were people asked what group they thought they were in?

Author Response

please see reviewer replies attached

Reviewer 2 Report

In this manuscript, the authors detail a clinical trial where they compare iTBS of the cerebellum combined with motor training with a sham condition. They use a modified randomization to ensure balanced groups given the small sample size. They show the iTBS plus motor training improved dystonia severity measured by TWSTRS and also quality of life was improved. The study is well designed and well reported.  I have two minor concerns below.

I am curious about the rationale for combing the iTBS with motor training rather than as iTBS alone as it appears that there was no independent effect of the motor training in the sham group.

The pegboard data is interesting.  It would be nice to expand on the possible explanations for the improved motor function in the hand.  Also it would be helpful to put the CD patients hand function in perspective. Is there performance on the pegboard test "worse" than subjects without dystonia. Is there "normalization" of hand function with the iTBS? 

Author Response

We thank the reviewer for their positive comments regarding our paper. We have addressed the concerns as outlined below.

 I am curious about the rationale for combing the iTBS with motor training rather than as iTBS alone as it appears that there was no independent effect of the motor training in the sham group.

We agree this is an interesting finding. We used a rationale borrowed from the stroke literature which considers non-invasive brain stimulation as 'priming' to make the brain more plastic prior to  motor training. This is considered to reduced the time and number of repetitions necessary when practising a motor task to evoke medium and long term changes in the brain. We added the sham iTBS to test the effect of the training alone. However, it is not overly surprising that motor training alone did not improve dystonia as there are many exercise based studies that also show no effect (eg Boyce et al 2013). The next step could be to see if exercise is needed at all or if iTBS produces the beneficial effect alone. see revised manuscript line 285 A study is warranted to examine effects of iTBS alone.

2. The pegboard data is interesting.  It would be nice to expand on the possible explanations for the improved motor function in the hand.  Also it would be helpful to put the CD patients hand function in perspective. Is there performance on the pegboard test "worse" than subjects without dystonia. Is there "normalization" of hand function with the iTBS? 

We agree. We speculate that neuromodulation of the cerebellum resulted in improved hand coordination. Unfortunately we did not include a control group so it is impossible to tell if our patients hand function was normalised. Comparison to normative values provided by Lafayette instruments is difficult as they separate data for sex and age range. Females non preferred hand -ages 40-54 mean 69.6 and SD 6.5 ages 55-70 mean 84.3 SD 15.3. Our pre-iTBS average was 84 and post was 67. The small numbers in our study would make separation into age brackets unreliable.  We agree this is an interesting discussion point and have added to our discussion, please see revised manuscript line 288 '..which improved coordination and fine motor control of the hand. It is well known the cerebellum projects to the Red Nucleus and Reticular Nucleus in the brainstem as well as to M1. All three descending pathways have terminations onto cervical motoneurons to control upper limb function. Without comparable data from control subjects we cannot tell if there were deficits in hand dexterity in people with cervical dystonia at baseline however, this preliminary study suggests cerebellar neuromodulation improves upper limb motor control in addition to dystonia severity. 

Round 2

Reviewer 1 Report

These edits satisfy my concerns.

Reviewer 2 Report

The authors have revised sufficiently.